# Elevated Intracranial Pressure in Cryptococcal Meningoencephalitis: Examining Old, New, and Promising Drug Therapies

**DOI:** 10.3390/pathogens11070783

**Published:** 2022-07-10

**Authors:** Abdulaziz H. Alanazi, Mir S. Adil, Xiaorong Lin, Daniel B. Chastain, Andrés F. Henao-Martínez, Carlos Franco-Paredes, Payaningal R. Somanath

**Affiliations:** 1Clinical and Experimental Therapeutics, College of Pharmacy, University of Georgia, Augusta, GA 30902, USA; aalanazi@augusta.edu (A.H.A.); miradil@stanford.edu (M.S.A.); 2Research Department, Charlie Norwood VA Medical Center, Augusta, GA 30912, USA; 3Department of Microbiology, University of Georgia, Athens, GA 30602, USA; xiaorong.lin@uga.edu; 4Department of Clinical and Administrative Pharmacy, UGA College of Pharmacy, SWGA Clinical Campus, Phoebe Putney Memorial Hospital, Albany, GA 31701, USA; daniel.chastain@uga.edu; 5Division of Infectious Diseases, University of Colorado, Anschutz Medical Campus, Aurora, CO 80045, USA; andres.henaomartinez@cuanschutz.edu (A.F.H.-M.); carlos.franco-paredes@cuanschutz.edu (C.F.-P.); 6Hospital Infantil de México, Federico Gómez, Ciudad de México 06720, Mexico

**Keywords:** *Cryptococcus neoformans*, meningitis, intracranial pressure, blood-brain-barrier, edema

## Abstract

Despite the availability of effective antifungal therapy, cryptococcal meningoencephalitis (CM) remains associated with elevated mortality. The spectrum of symptoms associated with the central nervous system (CNS) cryptococcosis is directly caused by the high fungal burden in the subarachnoid space and the peri-endothelial space of the CNS vasculature, which results in intracranial hypertension (ICH). Management of intracranial pressure (ICP) through aggressive drainage of cerebrospinal fluid by lumbar puncture is associated with increased survival. Unfortunately, these procedures are invasive and require specialized skills and supplies that are not readily available in resource-limited settings that carry the highest burden of CM. The institution of pharmacologic therapies to reduce the production or increase the resorption of cerebrospinal fluid would likely improve clinical outcomes associated with ICH in patients with CM. Here, we discuss the potential role of multiple pharmacologic drug classes such as diuretics, corticosteroids, and antiepileptic agents used to decrease ICP in various neurological conditions as potential future therapies for CM.

## 1. Introduction

Cryptococcal meningoencephalitis (CM) is accountable for more than 220,000 infections and 181,000 deaths each year, including approximately 15% of AIDS-associated deaths worldwide [1,2,3]. Although advanced HIV infection is an important risk factor for CM, other risk factors are also crucial to consider, including—but not limited to—transplant, sarcoidosis, immunoglobulin disorders, cell-mediated immunodeficiencies, diabetes mellitus, use of several biologicals, immunosuppressive agents, including corticosteroids, and several hematologic malignancies [4,5].

*Cryptococcus* infects immunocompetent individuals but remains dormant until an opportunity arises when individuals become immunocompromised [6]. In humans with immunosuppression, *Cryptococcus* encounters limited resistance in its route of entry [7]. Once reaching the lung parenchyma, *Cryptococci* enter through the respiratory tract and into the bloodstream and the central nervous system (CNS) [8]. *Cryptococcus* takes a transcellular route to the cerebrospinal fluid (CSF), perivascular spaces, and brain parenchyma to cross the blood-brain-barrier (BBB) but without affecting the blood–CSF barrier at the choroid plexus and the integrity of the BBB [9]. Another potential mechanism proposed for entry into the CNS is a “Trojan horse”, whereby *Cryptococcus* enters hidden inside mononuclear cells [10]. *Cryptococci* likely enter through spaces at different segments of the microvasculature such as the pial arteriolar trajectories into the subarachnoid space by allowing yeast access into the CSF, through arteriolar penetration in the subpial space allowing fungi to cross into the perivascular spaces, and via parenchymal capillaries that facilitates *Cryptococci* entry into the brain parenchyma [11]. The current review aims to present a comprehensive narration of the potential mechanisms involved in CM-associated intracranial hypertension (ICH), as well as to identify compounds used for ICH in other neurological conditions that can be repurposed for CM-associated ICH.

## 2. Intracranial Hypertension in the Setting of CNS Cryptococcosis

Studies indicate that ICH develops in around 75% of CM patients [12,13,14,15], and is responsible for early mortality and cognitive sequelae [15,16]. Symptoms in CM patients such as lethargy, altered mentation, personality changes, and memory loss have been linked to increased intracranial pressure (ICP) [14], including advanced HIV patients potentially caused by CSF outflow obstruction [17]. This, however, has been challenged by other instances where these symptoms appear in CM patients with no changes in ICP [15]. The fungal burden in many patients has been measured to be more than a million yeasts per milliliter of CSF, with increased polysaccharide antigen titers correlating to degree of ICP [18], which is confirmed in a retrospective study in the African population [19]. The precise mechanism for ICH and its causal link to the symptoms remains to be identified. There is, however, evidence that the CSF outflow blockage has been caused by the deposits of shed capsular polysaccharides and capsulated yeasts at the arachnoid villi, perivascular spaces, and in the brain parenchyma [20].

The brain is susceptible to rapid increases in ICP because of its containment in a rigid cranium restricting any increase in volume due to edema or any imbalance in the CSF production, circulation, or clearance [17,21]. The expansion of one of the cerebrum, CSF, or intravascular blood is at the expense of a reduction of another component.

## 3. Significance of Increased ICP in CM

Increased ICP has been correlated with increased morbidity and mortality in CM [15,22]. The increase in ICP mainly results from the cryptococcal yeast cells and shed, undegradable capsule polysaccharides that swell absorbing water and cause a physical obstruction, leading to CSF resorption abnormalities [23]. The increased ICP is associated with the number of organisms and the number of capsule polysaccharides in the arachnoid granulations [23,24]. Around three-quarters of patients with CM are projected to maintain a CSF pressure of greater than 25 cm H_2_O, as the normal range is between 6 and 25 cmH_2_O [25]. Previous studies reveal that patients with increased ICP have lower short-term survival than subjects with baseline pressures of less than 25 cm H_2_O [15]. Increased ICP may manifest as headache, vomiting, papilledema, confusion, visual acuity loss, and cranial nerve palsies [26,27,28]. Although a higher incidence of headache and neurologic findings is related to raised CSF opening pressures, ICH can occur without overt symptoms [15]. Currently, all ICP monitoring techniques are invasive. The two gold-standard approaches, external ventricular drain (EVD) and intraparenchymal probe, share a risk of bleeding or infection [29,30,31]. Another method to determine ICP is a lumbar puncture (LP), which is painful, highly invasive, and requires appropriately trained staff [32]. The World Health Organization (WHO) recommends performing serial lumbar punctures to lower ICP and determine the frequency of CSF drainage based on the symptom (accessed on 1 May 2022) [33]. A lumbar puncture should be repeated within 24 h if the opening pressure cannot be determined initially. If the manometer is unavailable, around 20 mL of CSF is recommended to be eliminated [34]. A lumbar puncture procedure to reduce ICP in CM patients is not always successful because, at times, the procedure fails to remove 20 mL of fluid [35]. Interestingly, decompression of the CSF volume by ventricular drainage, and therapeutics such as acetazolamide (AZA), have been reported to alleviate the raised CSF pressure in several non-CM-related neurological conditions [36]. Further, medical techniques associated with hyperosmolar therapy using agents such as mannitol and hypertonic saline are considered an effective primary treatment for elevated ICP by draining water [37]. Although the effectiveness of the above drugs has not been examined in CM patients with increased ICP, the success of these treatments with ICP in other diseases such as idiopathic intracranial hypertension (IIH), which is high pressure around the brain causing vision changes and headaches, suggests that these treatments may be useful in the management of ICP in CM patients.

## 4. Lessons from the Preclinical Studies

Several preclinical studies performed in animal models of ICP [38] have provided important clues as to the likely benefits of several drugs to manage CM patients and characterized their mechanisms of action (Table 1 and Table 2). Two diuretics, AZA and furosemide, were investigated and compared for their efficacies in animal models in relieving ICP. Studies have revealed that AZA and furosemide have comparable efficacies in rabbits [39,40] with AZA causing a reduction in ICP and modulation of the CSF secretion pathway [41]. Another study in healthy rats found no significant reduction in ICP on AZA [42]. Such discrepancies in results by various groups demand further preclinical research determining the potential beneficial effects of AZA for reducing ICP in CM and identifying associated molecular mechanisms.

Whereas ICP treatment in healthy rats using clinically relevant and at higher furosemide found no significant effect [43], a study conducted on dogs revealed a slight reduction in ICP generated by extradural mass lesion by furosemide [44]. Furosemide administration in newborn, preterm, and term rabbit pups substantially lowered ICP and led to a prominent decrease in CSF formation [45]. These effects might be attributable to the diuretic action of the therapy and/or hindrance to the production of CSF. A combination of furosemide and mannitol at 4-8-fold higher than the clinical doses caused a robust reduction in ICP in healthy dogs [46], and reduced brain water volume compared to only mannitol [43], possibly due to increased plasma osmolality.

Amiloride, a potassium-sparing diuretic, was effective for lowering elevated ICP in experimental brain edema in rats [47,48], but not in healthy rats [42], suggesting that the drug could be beneficial as adjunctive therapy in the treatment of ICH. We believe that the diverse outcomes between the above two studies could be due to the dissimilarities in the models that were used.

Because glucagon-like peptide-1 receptor (GLP-1R) agonists influence fluid homeostasis, some have attempted to repurpose these agents to treat ICP in CM patients. Intraperitoneally administered exendin-4 could modulate the production of cerebrospinal fluid (CSF) at the choroid plexus (CP) and hence lower ICP [49]. Accordingly, exendin-4 reduced ICP in normal and hydrocephalic rats, suggesting that this drug could be repurposed to manage ICH [50]. Likewise, subcutaneously administered 200 μg/kg Liraglutide, reduced cerebral edema in peri-contusional regions in a traumatic brain injury (TBI) rat model [49].

Although preclinical studies have revealed the mechanisms of the disease progression, how various treatments work mechanistically, and what outcomes can be expected in their use to treat ICP in various neurological conditions, none of the studies were performed in CM patients. Therefore, the information cannot be directly extrapolated to develop treatments for CM-associated ICP in patients. Nevertheless, because of the similarities in the development of ICP in various neurological diseases, the results provide reasonable optimism that some, if not all of the drugs investigated in the preclinical models may potentially be developed for the treatment of ICP in CM patients.

## 5. Pharmacological Considerations in the Treatment of CM

Treating CM is especially challenging as the distribution of therapeutics into the brain tissues is restricted by BBB and the lowering of drug concentration in the CNS by efflux pumps [12]. In addition, few antifungal agents are available to treat cryptococcosis, with amphotericin B deoxycholate (or its liposomal forms) with or without flucytosine used in induction therapy followed by long-term fluconazole maintenance therapy [51]. In addition to the limitation in the treatment options, *C. neoformans* can become resistant to flucytosine and fluconazole [52,53].

The management of CM comprises three antifungal therapy phases, which are “induction”, “consolidation”, and “maintenance” treatment [17]. Sterilization of CSF is the goal of the induction antifungal therapy, since a lower rate of fungal clearance is associated with a high number of deaths in the second and tenth weeks [54]. For the induction therapy, the WHO guideline recommends intravenous administration of 1.0 mg/kg/day of amphotericin B for one week along with 100 mg/kg/day of flucytosine, followed by fluconazole at a high dose (1200 mg/day) for seven days for the treatment of CM [33,55]. The combination of amphotericin B and flucytosine demonstrated enhanced survival against amphotericin B alone in a landmark trial [56]. For the consolidation phase, fluconazole at a modest dose (800 mg/day) for 8 weeks is recommended [33]. After that, patients will continue fluconazole at a lower dose (200 mg daily) for maintenance therapy for at least one year depending on immune reconstitution [33,51]. Long-term usage of fluconazole is beneficial, preventing relapse [57].

## 6. Clinical Studies on the Pharmacologic Management of Elevated ICP in CM and Non-CM Patients

Several clinical trials have been performed on the pharmacological management of ICP in patients with various neurological disorders (Table 3). Most of these studies, however, are performed in non-CM patients. Whether or not these treatments will be effective in CM patients is yet to be determined. Whereas some of these clinical trials were not successful in patients, refined formulations and conditions may be potentially helpful for treating CM patients.

AZA is widely used in the clinical setting to manage IIH [69], which is suggested to reduce CSF production, thus reducing the elevated ICP in IIH patients [69]. AZA reversibly inhibits water conduction using aquaporin-4 (AQP4), which has been implicated in cytotoxic brain edema resulting from water intoxication, brain ischemia, or meningitis [61]. The efficacy of AZA was addressed in a multicenter, randomized, double-masked, placebo-controlled study in subjects undergoing IIH and mild visual loss [70,71]. However, this trial did not incorporate patients with high ICP caused by CM. The outcome of this trial revealed that in conjunction with a low-sodium weight-reduction diet, usage of AZA moderately improves visual field function [71]. Moreover, AZA was found to improve the quality-of-life outcomes at six months [70]. However, its use is associated with common side effects, including loss of taste, nausea, and tingling in hands and feet [69].

A randomized controlled trial conducted on the use of AZA in IIH but not in CM patients failed due to insufficient sample size, as 48% of the subjects discontinued the drug for its side effects [76]. Coinciding with the above reports, a retrospective cohort study with AZA as adjunctive therapy to the standard therapeutics also showed no added advantage of AZA in reducing elevated CSF pressure in children between 3 to 15 years of age with non-cryptococcus-related meningitis [73]. A study on the use of AZA in treating ICH in CM patients was prematurely discontinued for safety reasons [72]. This randomized, double-blinded, placebo-controlled trial in patients with HIV diagnosed with CM associated with headache and CSF opening pressure of more than 20 cm H_2_O was linked to hyperchloremic acidosis, and several severe side effects that may have been complicated by deoxycholate amphotericin B that was co-administered [72].

Like AZA, Furosemide is suggested to lower ICH by reducing water volume and consequently alleviating the brain bulk in neurosurgical subjects [37,59]. Mechanistically, Furosemide increased serum osmolarity and decreased the water content in the brain [58]. Furosemide inhibits the sodium-potassium-2 chloride (Na+-K+-2 Cl−) symporter located in the ascending loop of the renal tubule [77]. The single-center retrospective study by Li et al. investigated the efficacy and safety of Furosemide as a supportive therapy with micro-pump infusions of 3% hypertonic saline (HS) in non-CM neurosurgical patients [37]. Continuous infusion of 3% HS with furosemide was found to be safe and effective for controlling ICP in neurological subjects with >25 cm H_2_O [37].

Topiramate, a sulfamate-substituted monosaccharide, has demonstrated a significant decrease in ICP in a preclinical study at both low and high clinically related doses by 32% and 21%, respectively [42]. Because of its ability to increase GABA activity and inhibit glutamate activity, Topiramate is used to treat patients with migraines and prevent seizures [65]. Topiramate also blocks voltage-gated sodium channels [78] and has demonstrated an effect in IIH reduction in two case reports, which prompted investigators to conduct an uncontrolled open-label study for IIH comparing the efficacy of topiramate to AZA [62,63,69]. In this trial, topiramate was found to be well tolerated and effective for managing IIH by reducing the production of CSF and decreasing body weight, particularly in obese patients [69]. Nevertheless, some patients receiving topiramate complained of distal paresthesia, concentration difficulties, and weight loss [69]. The above studies, however, did not include patients with CM; hence, the efficacy of Topiramate in alleviating ICP in CM patients is yet to be determined.

Mannitol has been retrospectively evaluated in patients living with HIV who had CM [67]. Mechanistically, mannitol promotes diuresis by increasing the concentration of filtrates in the kidneys and blocking the reabsorption of water by the kidney tubules [79]. The trial demonstrated that mannitol alleviated neurologic symptoms, particularly headaches associated with high ICP in CM [67]. Mannitol can induce the plasma to be relatively hypertonic, leading to an osmotic gradient from the CSF to the blood and then decreasing the high ICP [15] in CM patients. Although mannitol is the most commonly used hyperosmolar agent for the treatment of ICP [80], it is not used routinely, possibly due to the excessive dehydration, electrolyte abnormalities caused by the shift of free water into the intravascular space, and the possibility of crystal formation at low temperature [81].

Hypertonic saline (HS) demonstrated potential in lowering ICP in concentrations ranging from 3% to 23.4% in TBI patients [82] because of its reduced penetrability to the BBB, thereby drawing water from the CSF out of the cranium [83]. Therefore, it has been used as a gold standard for treating high ICP in various neurological diseases [74,84,85,86], including in children [68]. However, some adverse effects such as acute heart or kidney failure, severe pulmonary edema, and myelinolysis can appear due to receiving HS [66]. The efficacy of HS therapy in relieving ICP in CM patients is yet to be determined.

Corticosteroids are anti-inflammatory drugs that can suppress inflammation and edema caused by infection and hence are suggested for bacterial meningitis [87]. Dexamethasone has been assessed for its safety profile in patients with CM [75]. The study that was conducted on patients with HIV who had CM to determine whether adjunctive treatment with dexamethasone would improve patient survival reported that frequent use of dexamethasone was associated with a significantly higher risk of death or disability than placebo. Nevertheless, Dexamethasone was associated with a larger reduction in CSF opening pressure during the first 2 weeks than placebo. Since there was no disparity in mortality rate or Immune Reconstitution Inflammatory Syndrome (IRIS) determined among the two groups at 10 weeks, the study was stopped prematurely. Moreover, risks of adverse events and disability were higher and linked to the usage of dexamethasone in these patients [75].

Studies that are focused on *Cryptococcus*-induced changes in the endothelium and BBB contributing to brain edema are limited. As of today, most BBB studies in CM patients are focused on the route of infection [25,88,89,90,91,92]. Whether or not these contribute to brain edema is unclear. Accordingly, there is a critical need to understand the mechanistic pathway for CM at the BBB and vascular levels contributing to increased ICP and cerebral edema in CM patients.

## 7. Aquaporins (AQPs) as the Novel Therapeutic Targets for Treating Increased ICP

Aquaporins (AQPs), a family of membrane water channels, are expressed in various organs [93,94]. AQPs play essential roles in promoting passive water transport to cells [94]. Thirteen AQPs have already been identified in humans, including AQP1 and AQP4, which are extensively expressed in the CNS and thus appear to be a potential and attractive pharmaceutical target for many neurological diseases [95]. AQP1 has an essential function in the water transportation across the choroid plexus (CP) epithelial cells [96]. Since the apical membrane of the epithelial cells is extensively enriched with AQP1, its involvement in CSF production is anticipated [97]. A direct link could not be established for many years until a study revealed that AQP1^-/-^ mice had significantly impaired CSF secretion and maintained lower ICP than wild-type mice [60]. The study proposes that novel therapeutic options for decreasing high ICP could be achieved via AQP1 inhibition.

AQP4 is the primary water channel in the brain and is abundantly expressed in astrocytes in the CNS [94,95]. Astrocytic foot processes alongside the microvessels at the BBB are another area that is predominantly rich in AQP4 expression [96]. Expression modulation of AQP4 may affect the rate of edema formation [94]. AQP1^-/-^ mice had significantly reduced infarct size and edema in response to transient focal cerebral ischemia [98]. Contradictory to the finding of the beneficial effect of targeting AQP4 in stroke, another study showed that AQP4 knockout mice had higher mortality and exacerbated neurological impairments, implying that AQP4 contributes to the pathogenesis of stroke [99]. AQP4 may possess dual functions which are harmful in the development of edema and beneficial in resolving edema due to water removal [98,99]. A study on brain edema in AQP4-deficient mice after a focal ischemic stroke induced by middle cerebral artery occlusion illustrated that AQP4 deletion improved neurological outcomes [100]. More importantly, cerebral edema in these mice was lowered by 35% compared to wild-type (AQP4^+/+^) mice. Although these studies have not utilized the preclinical models of CM-induced ICP, the results from AQP4-deficient mice and stroke models suggest the potential utility of targeting AQP4 to reduce ICP in CM patients as well.

In the *Streptococcus pneumoniae* meningitis model, deletion of AQP4 produced a significant reduction in ICP and brain water accumulation in mice, suggesting that improved clinical outcomes could be achieved via blockage of AQP4 function [101]. Another study revealed increased ICP and death on intraperitoneal water-injection-induced severe water intoxication in transgenic mice overexpressing AQP4 specifically in glial cells [102]. Surprisingly, AQP4 deletion also led to elevated ICP along with higher water volume in the brain in a brain abscess model in mice [103]. Investigation of the complex interactions and temporal dynamics of AQPs would be important to understand these contradictory findings, and will be not only beneficial for the treatment of stroke and TBI, but also to treat patients suffering from ICP because of CM.

It is important to note that despite the recent advances in AQP4 research in CM, this is still in a conceptual stage. Because AZA has been demonstrated to enhance AQP4 in astrocytes, AZA may have a role in pharmacologically modulating AQP4 expression in astrocytes in CM patients experiencing increased ICP. However, the potential risks of nephrotoxicity and electrolyte disturbances, associated with both AZA and antifungal therapies, will need to be minimized in future clinical trials.

## 8. Conclusions

The clinical and animal studies indicated diverse consequences of several drugs discussed in the current review. The best way to identify a successful pharmacological approach in the management of raised ICP caused by CM is by studying the effect of various drugs at the molecular level, followed by evaluating outcomes at a clinical-stage in the process of the bench to bedside transition. There is also a need for drug repurposing if the potential of the current therapeutic options is uncertain despite modulating the dose.

## Figures and Tables

**Table 1 pathogens-11-00783-t001:** Compounds and their mechanisms of action in reducing intracranial pressure.

Drug	Structure (PubChem)	Mechanism of Action	Ref.
Furosemide CAS: CAS 54-31-9 MF: C_12_H_11_ClN_2_O_5_S MW: 222.3 g/mol	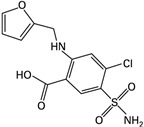	Increases serum osmolality and decreases water content in the brain. Inhibits the sodium-potassium-2 chloride (Na+-K+-2 Cl−) symporter located in the ascending loop of the renal tubule.	[39,43,44,45,58]
Amiloride CAS: 2609-46-3 MF: C_6_H_8_ClN_7_O MW: 330.74 g/mL	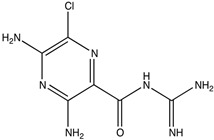	Potassium-sparing diuretic.	[47]
Acetazolamide (CAS: 59-66-5) MF: C_4_H_6_N_4_O_3_S_2_ MW: 229.63 g/mol	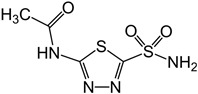	Inhibits carbonic anhydrase activity. Reversible inhibitor of water conduction by aquaporin-4.	[40,41,59,60,61]
Topiramate CAS: 97240-79-4 MF: C_12_H_21_NO_8_S MW: 339.36 g/mol	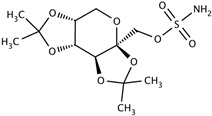	Increases GABA and inhibits glutamate activities.Inhibits carbonic anhydrase activity. Blocks voltage-gated sodium channels.	[38,62,63,64,65]
Exendin-4 CAS: 141758-74-9 MF: C_184_H_282_N_50_O_60_S MW: 4187 g/mol	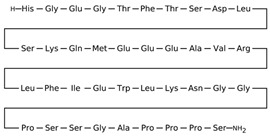	Glucagon-like peptide-1 receptor agonist that modulates cerebrospinal fluid secretion at the choroid plexus and subsequently reduces ICP. Reduced Na+- and K+-dependent adenosine triphosphatase activity, a key regulator of CSF secretion.	[50]
Liraglutide CAS: 204656-20-2 MF: C_172_H_265_N_43_O_51_ MW: 3751 g/mol	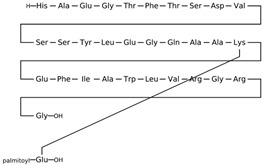	Glucagon-like peptide-1 receptor agonist, which, like Exendin-4, modulates cerebrospinal fluid secretion at the choroid plexus and subsequently reduces ICP.	[49]
Mannitol CAS: 69-65-8 MF: C_6_H_14_O_6_ MW: 182.17 g/mol	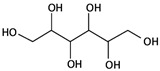	Increases plasma osmolality that reduces water content in the brain.	[43,46,66,67,68]

**Table 2 pathogens-11-00783-t002:** Pre-clinical studies on compounds targeting ICP in experimental animal models.

Drug	Animal	Effect		
		Healthy Animals	Disease Model	Ref.
Furosemide	Rats	No effect on intracranial pressure (ICP) with a clinically relevant dose but a high dose reduces brain volume.		[43]
	Dogs	Robust reduction in ICP on a high dose.	A slight reduction in ICP.	[44]
	Rabbits		Prominent reduction in ICP.	[40]
Amiloride	Rats	No change in ICP.	Lowers elevated ICP.	[42]
Acetazolamide	Rats	*Mixed response:* While one study revealed a reduction in ICP at a 200 mg dose, another study reported no change despite administering high doses (oral or subcutaneous).	55% reduction in cerebrospinal fluid production.	[41,42,60]
Topiramate	Rats	Significant decrease in ICP at low and high clinically related doses by 32% and 21%, respectively.		[42,65]
Exendin-4	Rats	Reduced ICP.	Reduced ICP in hydrocephalus	[50]
Liraglutide	Rats		Reduced cerebral edema in peri-contusional regions in the traumatic brain injury (TBI) at a dose of 200 μg/kg.	[49]

**Table 3 pathogens-11-00783-t003:** Clinical trials on treatments targeting intra-cranial pressure in patients with neurological diseases.

Drug	Study	Population	Outcome	Adverse Effects	Ref.
Acetazolamide	Multicenter, randomized, double-masked, placebo-controlled trial.	Subjects undergoing Idiopathic intracranial hypertension (IIH) and mild visual loss.	In combination with a low-sodium weight-reduction diet, it moderately improved visual field function. It also improved the quality-of-life outcomes at six months.	Changed taste, nausea, fatigue, and tingling of the hands and feet.	[38]
	A randomized, double-blinded, placebo-controlled trial.	Adults with HIV + Cryptococcus Meningoencephalitis (CM) + headache + >20 cm H_2_O CSF opening pressure.	Discontinued for safety reasons.	Electrolyte imbalance, particularly in bicarbonate and chloride levels, often hyperchloremic acidosis.	[72]
	A randomized controlled trial.	IIH but not CM.	Failed due to insufficient sample size.	48% of the subjects discontinued the drug for its side effects.	[61]
	Retrospective cohort study.	3–15-year-old children with non-Cryptococcus-related meningitis.	AZA as adjunctive therapy to the standard therapeutics showed no added advantage of AZA in reducing elevated CSF.		[73]
Furosemide	Single-center retrospective study.	Non-CM neurological subjects with >25 cm H_2_O.	Continuous infusion of 3% HS with Furosemide was safe and effective in controlling ICP.		[37]
Topiramate	Uncontrolled open-label study.	Non-CM patients with IIH.	It was effective and well-tolerated for the management of IIH.	Distal paresthesia and concentration difficulties, in addition to weight loss.	[69]
Hypertonic saline		Non-CM patients with traumatic brain injury (TBI).	Effective at lowering ICP in concentrations ranging from 3% to 23.4%.	Acute heart or kidney failure, severe pulmonary edema, and myelinolysis.	[74]
Dexamethasone	A randomized controlled trial.	CM patients.	The study was stopped prematurely for no disparity in mortality rate or immune reconstitution inflammatory syndrome (IRIS) among two groups at 10 weeks.	Risks of disability and clinical adverse events were higher.	[75]

## Data Availability

Not applicable.

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
