# Peer review of "Elevated Intracranial Pressure in Cryptococcal Meningoencephalitis: Examining Old, New, and Promising Drug Therapies"

_pathogens, 2022, doi:10.3390/pathogens11070783_

Round 1

Reviewer 1 Report

The manuscript is interesting and presents the roles of multiple pharmacologic classes: diuretics, corticosteroids, antiepileptic drugs for decreasing ICP in various neurological conditions and potential future therapies for Cryptococcal meningoencephalitis, including aquaporin 4 (AQP4) transporter. I suggest minor corrections: including one column in each of the tables (indicating the references), name of drugs in lower case. Besides, for WHO site - to mention the accession date etc. All these suggestions are marked in the documents attached.

Author Response

We thank the reviewer for the constructive critique. We have addressed the concerns raised and have incorporated the suggested changes in this revised version. We believe that these changes have helped improve the quality of the manuscript. Our point-by-point response to reviewer comments is delineated below:

General comment: The manuscript is interesting and presents the roles of multiple pharmacologic classes: diuretics, corticosteroids, antiepileptic drugs for decreasing ICP in various neurological conditions, and potential future therapies for Cryptococcal meningoencephalitis, including aquaporin 4 (AQP4) transporter.

Response: We thank the reviewer for the positive comments on the manuscript.  

Critique 1: I suggest minor corrections: including one column in each of the tables (indicating the references), and the name of drugs in lower case.

Response: We thank the reviewer for this suggestion. We have included references in the tables.  

Critique 2: Besides, for the WHO site - to mention the accession date, etc. All these suggestions are marked in the documents attached.

Response: We thank the reviewer for the careful review and for identifying these minor errors. We have incorporated all the suggested changes.

Reviewer 2 Report

Manuscript is well written and has  highlighted the importance  of decreasing the ICP other than the standard treatment for management of cryptococcosis. But it is to lengthy needs to be shortened.

Author Response

We thank the reviewer for the constructive critique. We have addressed the concerns raised and have incorporated the suggested changes in this revised version. We believe that these changes have helped improve the quality of the manuscript. Our point-by-point response to reviewer comments is delineated below:

General comment: Manuscript is well written and has highlighted the importance of decreasing the ICP other than the standard treatment for the management of cryptococcosis.

Response: The authors appreciate the positive comments.

Critique 1: But it is too lengthy and needs to be shortened.

Response: We have shortened several sections in the manuscript, mainly the pre-clinical and endothelial/BBB maintenance sections. We have also shortened the introduction section.

Reviewer 3 Report

In this review the authors take a look at a rather interdisciplinary area in the field of cryptococcosis, that being the clinical management of elevated intracranial pressure. They offer a good introduction of the basic elements for a less clinical audience, and provide a detailed discussion of some front line agents for management of elevated intracranial pressure. They finish with a section on the promising area of aquaporins as therapeutic targets.

On the whole, the level of detail and structure is good. I have two major and a few minor points to raise:

1. some of the content on preclinical studies becomes a little redundant in light of the subsequent findings in human studies. A good attempt is made at establishing some of the basic science of the problem, but several parts of these sections are repetitive (for example 123-124 and 218 to 220--it seems redundant to discuss the mouse studies in such detail when so much human data is available).

2. In total there are very few actual human studies of specific treatments for elevated intracranial pressure specifically in the context of cryptococcal meningoencephalitis. Multiple times it is pointed out that a particular approach has not been tested specifically in this context but there is cause for optimism. This would better be stated as a general theme. The state of research is well away from infection-specific targeting approach, but the aquaporin work in S pneumoniae brings up the possibility of this in cryptococcal infection. This is the most inspiring part of the review but it is tucked in at the end instead of serving as a launching point for the discussion.

More specific points:

-lines 47-48. the assertion that BBB crossing takes place without affecting the integrity of the BBB is controversial, and is even contradicted later in this piece (line 203). The entire subject of mechanisms of BBB crossing seems tangential to the aim of this paper: those events took place long before the pathology of interest (elevated intracranial pressure) began.

Lines 54-58: again confusing the final pathology (including a paucity of inflammation in the subarachnoid space and meninges) with the early events that led to it, the anatomical pathology of which we know practically nothing about.

Lines 79-80. This sentence is probably best left out, as the relationship between mass brain lesions and clinical manifestations is actually much more complex and what is stated doesn't add to the review.

Paragraph starting at line 199--again a recounting of studies regarding BBB crossing, which chronologically, and probably pathogenically, far removed from the focus of the manuscript.

line 333 spelling: Streptococcus pneumoniae

Author Response

We thank the reviewer for the constructive critique. We have addressed the concerns raised and have incorporated the suggested changes in this revised version. Based on the reviewers’ ‘thematic’ suggestion, we have changed the title of the manuscript to “Elevated intracranial pressure in Cryptococcal meningoencephalitis: Examining old, new, and promising drug therapies”, We believe that these changes have helped improve the quality of the manuscript. Our point-by-point response to reviewer comments is delineated below:

General comment: In this review, the authors take a look at a rather interdisciplinary area in the field of cryptococcosis, that being the clinical management of elevated intracranial pressure. They offer a good introduction of the basic elements for a less clinical audience and provide a detailed discussion of some front-line agents for the management of elevated intracranial pressure. They finish with a section on the promising area of aquaporins as therapeutic targets. On the whole, the level of detail and structure is good.

Response: The authors appreciate the positive comments.

Critique 1: Some of the content on preclinical studies becomes a little redundant in light of the subsequent findings in human studies. A good attempt is made at establishing some of the basic science of the problem, but several parts of these sections are repetitive (for example 123-124 and 218 to 220--it seems redundant to discuss the mouse studies in such detail when so much human data is available).

Response: We thank the reviewer for this excellent suggestion. We have removed the sentences that are repetitive and reduced the descriptions on the mouse data.

Critique 2: In total there are very few actual human studies of specific treatments for elevated intracranial pressure specifically in the context of cryptococcal meningoencephalitis. Multiple times it is pointed out that a particular approach has not been tested specifically in this context but there is cause for optimism. This would better be stated as a general theme. The state of research is well away from an infection-specific targeting approach, but the aquaporin work in S pneumoniae brings up the possibility of this in cryptococcal infection. This is the most inspiring part of the review but it is tucked in at the end instead of serving as a launching point for the discussion.

Response: We agree with the reviewer to focus the discussions on a general theme of targeting CM patients with ICP, and how pharmacological modulation of AQP4 expression is a new concept to treat CM patients with ICP. We have also changed the title of the article accordingly. We have also removed the repetitive statements for specific approaches and removed much of the discussions on BBB regulation. This also helps us shorten the manuscript as suggested by reviewer 2.

Critique 3: Lines 47-48. The assertion that BBB crossing takes place without affecting the integrity of the BBB is controversial and is even contradicted later in this piece (line 203). The entire subject of mechanisms of BBB crossing seems tangential to the aim of this paper: those events took place long before the pathology of interest (elevated intracranial pressure) began.

Response: We agree with the reviewer that there is so much discrepancy in the literature over the mechanisms of BBB crossing of Cn. The discrepancy could be at least partly attributed to the different time points when BBB integrity was examined. The idea of including these reports, despite being paradoxical at times, is to discuss the literature and present the best possible scenario on BBB crossing by Cn. We have shortened this section to make it less distracting from the main message of the manuscript

Critique 4: Lines 54-58: again confusing the final pathology (including a paucity of inflammation in the subarachnoid space and meninges) with the early events that led to it, the anatomical pathology of which we know practically nothing about.

Response: We have removed this from the article.

Critique 5: Lines 79-80. This sentence is probably best left out, as the relationship between mass brain lesions and clinical manifestations is actually much more complex and what is stated doesn't add to the review.

Response: We have removed the suggested sentence from the article.

Critique 6: Paragraph starting at line 199--again a recounting of studies regarding BBB crossing, which chronologically, and probably pathogenically, far removed from the focus of the manuscript.

Response: We have removed this information from the article and mentioned it as a gap in knowledge and a topic for future investigations.

Critique 7: Line 333 spelling: Streptococcus pneumoniae

Response: We apologize for this error. This has been corrected.

Reviewer 4 Report

 This paper reviews the topic of increased intrcranial pressure (ICP)  in cryptococcal meningitis. Some features of the  pathology and pathogenesis of cryptococcosis are included, as well as a brief discussison about five therapeutic agents that could be used.

The authors begin with the assumption that symptoms of cryptococcal meningitis are “mostly” caused by increased ICP (lines 64-69). This fails to account for the same symptoms in the 50-60% who don’t have increased ICP and the failure of ventriculoperitoneal shunting to improve most of the symptoms. Shunting for ICP in cryptococcal meningitis is thought to improve decreased visual acuity and decrease mortality but not decrease other symptoms. Shunting for hydrocephalus due to cryptococcal meningitis, which usually does not include elevated ICP, can improve cognition and gait, but not headache, nausea, and cranial nerve palsies, including deafness. Therefore, the goals of reducing intracranial pressure are less than the authors believe.

The authors concepts of CSF flow are out of date. I recommend Prouix’s review in Cellular and Molecular Life Sciences 2021;  78:242-2457.

The concept that removing 20 ml of CSF can reduce ICP (line 109) fails to account for the production on 20ml per hour in the average individual.

The five therapeutic agents for elevated ICP chosen for review include two that were not found useful in randomized clinical trials (acetazolamide and dexamethasone) and three that are not recommended by anyone.

The 30-40 lines about aquaphorin 4 don’t lead to any management suggestions and are theoretical at present.

Author Response

We thank the reviewer for the constructive critique. We have addressed the concerns raised and have incorporated the suggested changes in this revised version. We believe that these changes have helped improve the quality of the manuscript. Our point-by-point response to reviewer comments is delineated below:

Critique 1: The authors begin with the assumption that symptoms of cryptococcal meningitis are “mostly” caused by increased ICP (lines 64-69). This fails to account for the same symptoms in the 50-60% who don’t have increased ICP and the failure of ventricular-peritoneal shunting to improve most of the symptoms. Shunting for ICP in cryptococcal meningitis is thought to improve decreased visual acuity and decrease mortality but not decrease other symptoms. Shunting for hydrocephalus due to cryptococcal meningitis, which usually does not include elevated ICP, can improve cognition and gait, but no headache, nausea, and cranial nerve palsies, including deafness. Therefore, the goals of reducing intracranial pressure are less than the authors believe.

Response: We agree with the reviewer that ICP is not developed in all CM patients, and some of the behavioral symptoms can be observed in CM patients with no ICP, and patients with ICP can be non-symptomatic. Nevertheless, managing ICP in CM patients may still have benefits in improving the quality of life in CM patients, although that alone may not be the right approach to disease management. We have reduced the emphasis on ICP association with symptoms and revised the section accordingly.

Critique 2: The author's concepts of CSF flow are out of date. I recommend Prouix’s review in Cellular and Molecular Life Sciences 2021;  78:242-2457. The concept that removing 20 ml of CSF can reduce ICP (line 109) fails to account for the production of 20ml per hour in the average individual.

Response: We agree. Please note that we are only stating the standard recommendation of removing 20 ml of CSF to manage ICP in some CM patients. Whether or not they are sufficient could be the point of debate that we have included in the revision based on the reviewer’s comments.

Critique 3: The five therapeutic agents for elevated ICP chosen for review include two that were not found useful in randomized clinical trials (acetazolamide and dexamethasone) and three that are not recommended by anyone.

Response: That is a great point. The theme of this review article is to discuss the data published about these agents and how each of these fared in the pre-clinical and clinical studies. It is correct that acetazolamide failed in one clinical trial for CM patients. However, this clinical trial was not without limitations, hence has scope for improvement and success, which have been discussed in the review article. For, other drugs, whether or not these will be useful for CM patients has not been tested. There could also be new drugs in the pipeline that could help in the future. We have also revised the title to provide more clarity on the theme of the article.

Critique 4: The 30-40 lines about aquaporin 4 don’t lead to any management suggestions and are theoretical at present.

Response: We agree with the reviewer that the AQP4 is currently in a conceptual stage for ICP management in CM patients. With acetazolamide demonstrated to have the ability to enhance AQP4 in astrocytes, changing the conditions of acetazolamide therapy would be one of the various ways AQP4 can be targeted to treat CM patients with ICP. We have made a note of this in the article emphasizing the need for further investigation.

Round 2

Reviewer 4 Report

meaning of sentence on lines 90-92 is unclear.

Author Response

Critique: meaning of the sentence on lines 90-92 is unclear.

Response: We apologize if the meaning of the abovementioned sentence was not clear from how we wrote it. The sentence explains that lumbar puncture is not an option for all patients at all times because occasionally it fails to remove 20 ml of fluid with no significant effect on reducing ICP. We have revised the sentence for more clarity.